# Unique Mode of Antiviral Action of a Marine Alkaloid against Ebola Virus and SARS-CoV-2

**DOI:** 10.3390/v14040816

**Published:** 2022-04-15

**Authors:** Mai Izumida, Osamu Kotani, Hideki Hayashi, Chris Smith, Tsutomu Fukuda, Koushirou Suga, Masatomo Iwao, Fumito Ishibashi, Hironori Sato, Yoshinao Kubo

**Affiliations:** 1Department of Clinical Medicine, Institute of Tropical Medicine, Nagasaki University, Nagasaki 852-8523, Japan; christopher.smith@lshtm.ac.uk; 2Pathogen Genomics Center, National Institute for Infectious Diseases, Tokyo 208-0011, Japan; konioo@niid.go.jp; 3Medical University Research Administrator, Nagasaki University School of Medicine, Nagasaki 852-8523, Japan; hhayashi@nagasaki-u.ac.jp; 4Department of Clinical Research, London School of Hygiene and Tropical Medicine, London WC1E 7HT, UK; 5School of Tropical Medicine and Global Health, Nagasaki University, Nagasaki 852-8523, Japan; 6Environmental Protection Center, Nagasaki University, Nagasaki 852-8131, Japan; t-fukuda@nagasaki-u.ac.jp; 7Graduate School of Engineering, Nagasaki University, Nagasaki 852-8131, Japan; iwao@nagasaki-u.ac.jp; 8Graduate School of Fisheries and Environmental Sciences, Nagasaki University, Nagasaki 852-8131, Japan; sugakisi@nagasaki-u.ac.jp (K.S.); fumito@nagasaki-u.ac.jp (F.I.); 9Organization for Marine Science and Technology, Nagasaki University, Nagasaki 852-8131, Japan; 10Program for Nurturing Global Leaders in Tropical and Emerging Communicable Diseases, Graduate School of Biomedical Sciences, Nagasaki University, Nagasaki 852-8523, Japan

**Keywords:** marine alkaloid, lamellarin α 20-sulfate, emerging viruses, antiviral action, antiviral mechanism, pseudotyped lentiviral vector, molecular dynamics simulation, docking simulation

## Abstract

Lamellarin α 20-sulfate is a cell-impenetrable marine alkaloid that can suppress infection that is mediated by the envelope glycoprotein of human immunodeficiency virus type 1. We explored the antiviral action and mechanisms of this alkaloid against emerging enveloped RNA viruses that use endocytosis for infection. The alkaloid inhibited the infection of retroviral vectors that had been pseudotyped with the envelope glycoprotein of Ebola virus and SARS-CoV-2. The antiviral effects of lamellarin were independent of the retrovirus Gag-Pol proteins. Interestingly, although heparin and dextran sulfate suppressed the cell attachment of vector particles, lamellarin did not. In silico structural analyses of the trimeric glycoprotein of the Ebola virus disclosed that the principal lamellarin-binding site is confined to a previously unappreciated cavity near the NPC1-binding site and fusion loop, whereas those for heparin and dextran sulfate were dispersed across the attachment and fusion subunits of the glycoproteins. Notably, lamellarin binding to this cavity was augmented under conditions where the pH was 5.0. These results suggest that the final action of the alkaloid against Ebola virus is specific to events following endocytosis, possibly during conformational glycoprotein changes in the acidic environment of endosomes. Our findings highlight the unique biological and physicochemical features of lamellarin α 20-sulfate and should lead to the further use of broadly reactive antivirals to explore the structural mechanisms of virus replication.

## 1. Introduction

Ebola virus is a negative single-stranded RNA virus that belongs to the filoviridae family. Ebola virus epidemics occur sporadically in Africa, and the Ebola virus disease is characterized by hemorrhagic fever, excessive inflammation, coagulopathy, and multi-organ failure with a high mortality rate. The largest Ebola virus outbreak caused almost 30,000 cases in Guinea, Liberia, and Sierra Leone from 2013 to 2016 [1]. An Ebola virus vaccine was approved by the FDA in 2019 and efficiently inhibited its spread. However, only a few therapeutic drugs against Ebola virus have been developed [2], and Ebola virus may become resistant to those drugs. Thus, additional anti-Ebola virus drugs need to be developed.

It has been reported that endocytosis, endosome acidification, and endosome cathepsin proteases inhibitors attenuate infection mediated by the viral envelope glycoproteins of the Ebola or SARS-CoV-2 virus [3,4,5,6]. These results indicate that these viral particles are internalized into the endosomes of host cells, following the binding to their cell surface receptors, and the envelope glycoproteins are then cleaved by cathepsin proteases activated by endosome acidification to potentiate the membrane fusion capability of the envelope glycoproteins for viral entry.

On the other hand, the replication-defective human immunodeficiency virus type 1 (HIV-1) vector is widely used as a tool to transfer a gene of interest into target cells. HIV-1 vector particles can incorporate distinct viral envelope glycoproteins, and such vectors are designated as pseudotyped lentiviral vectors [7,8]. Pseudotyped lentiviral vectors infect target cells via the mechanisms of the incorporated viral glycoproteins. Even though Ebola virus is highly pathogenic and thus must be used in BSL4 laboratories, replication-defective lentiviral vectors containing its glycoprotein can be manipulated in BSL2 experimental rooms. This pseudotyped HIV-1 vector is used to understand the molecular mechanisms of Ebola virus and SARS-CoV-2 virus infections and to identify inhibitory compounds against these viruses [4,9,10,11,12,13,14].

Many antiviral drugs have been isolated from marine organisms [15,16,17,18,19,20], and it has been reported that lamellarin α 20-sulfate, a marine alkaloid, suppresses HIV-1 infection through the inhibition of integrase [21,22]. However, this compound has a charged, hydrophilic sulfate residue that is required for its antiviral activity [23]. Therefore, little lamellarin α 20-sulfate should cross the cell membrane to the cytoplasm and gain access to the HIV-1 integrase on the inside of membrane-enveloped cells or virions. Instead, we have found that the compound attenuates the cell fusion activity of the HIV-1 envelope glycoprotein, but not the enzymatic activity of integrase [23].

Sulfated polysaccharides, including heparin, dextran sulfate, fucoidan, and chondroitin sulfate, also inhibit the replication of various viruses [24,25,26,27,28], and their sulfate residues are required for the antiviral activity [23,29,30,31,32], suggesting that lamellarin α 20-sulfate may inhibit HIV-1 infection by a mechanism that is similar to that of sulfated polysaccharides. In this study, we analyzed the effects of lamellarin α 20-sulfate and sulfated polysaccharides on pseudotyped lentiviral vector infection mediated by envelope glycoproteins of the Ebola virus to evaluate their potential antiviral activity. It has already been reported that heparin and chondroitin sulfate inhibit Ebola virus glycoprotein-mediated infection [28], but the impacts of dextran sulfate and fucoidan have not been analyzed yet. In addition, we characterized the sites and chemical features of molecular interactions between these sulfated compounds and the Ebola virus glycoprotein in order to better understand the molecular mechanisms of their antiviral actions. Our data illustrate a hitherto unappreciated mechanism in sulfated compounds—namely, the inhibition of conformational glycoprotein changes—that plays a role in the antiviral action of lamellarin α 20-sulfate against Ebola virus, whose mechanism of infection involves endocytosis.

## 2. Materials and Methods

### 2.1. Cells

Human 293T, HeLa, and TE671 cells were cultured in Dulbecco’s modified Eagle’s medium (Sigma-Aldrich) containing 8% fetal bovine serum and 1% penicillin–streptomycin (Sigma-Aldrich) at 37 °C with a 5% CO_2_ atmosphere, as previously described [33].

### 2.2. Chemicals

Lamellarin α 20-sulfate (CAS RN 896115-11-0) was synthesized in our laboratory as reported [23]. Heparin sodium salt from porcine intestinal mucosa (CAS RN 9041-08-1) and fucoida from *Macrocystis pyrifera* (CAS RN 9072-19-9) were purchased from Sigma-Aldrich. Chondroitin sulfate C sodium salt (CAS RN 12678-07-8) and dextran sulfate sodium salt 5000 (molecular weight 1000–9000. CAS RN 9011-18-1) were purchased from FUJIFILM Wako Pure Chemical Corporation.

### 2.3. Antibody

Neutralizing antibody against SARS-CoV-2 (40592-R001) [34] was purchased from Sino Biological Inc. Antiserum isolated from rabbit immunized with Ebola virus glycoprotein (GTX134144) [35] was obtained from Gene Tex.

### 2.4. Plasmids

The HIV-1 Gag-Pol-Tat-Rev expression plasmid (R8.91) was kindly provided by Dr. D. Trono [36]. The luciferase-encoding HIV-1 vector genome and VSV-G expression plasmids were obtained from the AIDS Research and Reference Reagent Program, NIAID, NIH, USA [37,38]. The SARS-CoV-2 spike protein expression plasmid (VG40589-UT) [39] was purchased from Sino Biological Inc. The Ebola virus glycoprotein expression plasmid was kindly provided by Dr. Kawaoka [40]. The MLV Gag-Pol expression plasmid [41] was purchased from TaKaRa Bio. The luciferase-encoding MLV vector genome expression plasmid was constructed in this study.

### 2.5. Preparation of Pseudotyped HIV-1 and MLV Vectors

To construct the pseudotyped HIV-1 vector, human 293T cells were transfected with the HIV-1 Gag-Pol-Tat-Rev (1 µg) and luciferase-encoding HIV-1 vector genome (1 µg) expression plasmids together with the viral glycoprotein expression plasmid (1 µg) using Fugene transfection reagent (5 µL) (Promega) in a 6 cm dish [33]. Culture supernatants of the transfected cells were changed to fresh media 24 h after the transfection, and then the cells were cultured for an additional 24 h. Culture supernatants from the transfected cells were centrifuged at 1,200 rpm for 5 min to remove the cells and were then inoculated to target cells. 

To construct a pseudotyped MLV vector, 293T cells were transfected with the MLV Gag-Pol (1 µg) and luciferase-encoding MLV vector genome (1 µg) expression plasmids together with the viral glycoprotein expression plasmid (1 µg) using Fugene transfection reagent (5 µL) in a 6 cm dish [41]. The culture supernatants of the transfected cells were changed to fresh media 24 h after the transfection, and then, the cells were cultured for an additional 24 h. Culture supernatants from the transfected cells were centrifuged at 1200× *g* rpm for 5 min to remove the cells, and they were then inoculated to target cells.

### 2.6. Treatment of Target Cells with Lamellarin α 20-Sulfate

Target cell treatment with lamellarin α 20-sulfate was essentially carried out as previously described [42]. Briefly, target 293T cells were treated with lamellarin α 20-sulfate in three-different ways as follows: (i) Lamellarin α 20-sulfate (150 µM) was added to the target 293T cells, and the cells were cultured for 5 h. The treated cells were washed with the medium three times to remove the compound, and they were then inoculated with the Ebola virus-pseudotyped HIV-1 vector. The inoculated cells were washed with the medium three times 5 h after the inoculation to remove the vector and were cultured for 2 days (pretreated). (Ii) Target 293T cells were inoculated with pseudotyped HIV-1 vector in the presence of lamellarin α 20-sulfate and were washed with the medium to remove the vector and compound. Then, the inoculated cells were cultured for 2 days (cotreated). (Iii) Target 293T cells were inoculated with the pseudotyped HIV-1 vector in the absence of the compound. The inoculated cells were washed with the medium three times 5 h after the inoculation to remove the vector and were treated with the compound for 5 h. The treated cells were washed with the medium three times to remove the compound and then were cultured for 2 days (post-treated).

### 2.7. Infectivity Assay

Cells inoculated with culture supernatants containing pseudotyped viral vector were cultured for 2 days. Cell lysates were prepared from the cells. The luciferase activities of the cell lysates were measured by a luciferase assay system (Promega) using a luminometer (Beckman Coulter LD400) to estimate the transduction titers.

### 2.8. Assessment of Cell Viability

Human 293T cells (10^6^ cells) were seeded into a 3 cm dish and cultured for 24 h. Lamellarin α 20-sulfate, heparin, chondroitin sulfate, fucoidan, or dextran sulfate was added to the culture, and the cells were cultured for 2 days. Cells were stained with trypan blue, and the unstained cells were counted.

### 2.9. Statistical Analysis

Differences between the two groups were determined by Student’s *t*-test. Differences were considered statistically significant if the *p*-value was <0.05 for all tests. 

### 2.10. Molecular Modeling of the Ebola Virus Glycoprotein Trimer in the Ligand-Free State

A structural model of the glycoprotein trimer of the Reston Ebola virus strain (GenBank accession no. AAC54885) [43] used in the pseudotyped virus study was constructed by homology modeling, followed by molecular dynamics (MD) simulation, as described for the modeling of trimeric envelope glycoproteins such as HIV-1 and the influenza virus [44,45,46]. Briefly, the reported X-ray crystal structure of the unliganded glycoprotein trimer of the Zaire Ebola virus strain at a resolution of 2.23 Å (PDB code: 5JQ3 [47]) was used as the modeling template for the homology modeling that was carried out with the Molecular Operating Environment (MOE) 2019.01 (Chemical Computing Group Inc., Montreal, QC, Canada). Gal_2_Man_3_GlcNAc_4_ was added to the model based on the information from the *N*-glycosylation sites [48] using the Online Glycoprotein Builder web-tool, GLYCAM-Web [49]. The obtained model was optimized via energy minimization using MOE and an Amber10: Extended Hückel Theory (EHT) force field implemented in MOE, which combines the Amber10- and EHT-bonded parameters for large-scale energy minimization [50]. Subsequently, the model was subjected to MD simulations to obtain a thermodynamically stable structure in solution. The simulation was performed using the pmemd module in the Amber 16 program package [51] with the *ff14SB* force field for the protein simulation [52] and with the *GLYCAM06j-1* force field for the glycan simulation [53]. The TIP3P water model [54] was used to simulate the aqueous solution. A non-bonded cutoff of 10 Å was used. Bond lengths involving hydrogen were constrained with SHAKE, a constraint algorithm to satisfy a Newtonian motion [55], and the trajectory data of all MD simulations were collected at 2 fs intervals. After heating to 310 K, 20 ps simulations were first carried out using the NVT ensemble followed by longer simulations executed using the NPT ensemble at 1 atm, 310 K, and in 150 mM NaCl. 

### 2.11. Molecular Patch Analysis

The interaction-prone areas on the Ebola virus glycoprotein were estimated using the “Protein Patch Analyzer” tool in MOE as previously described [56,57,58,59,60]. A timer model of the Ebola virus glycoprotein after 100 ns of MD simulation was used for the patch analyses. Briefly, the Protein Patch Analyzer tool was applied to search for the positively charged patches (a minimal patch area of 40 Å^2^ in size) that were potentially involved in interaction with negatively charged molecules. The tool was also applied to search for the hydrophobic patches (a minimal patch area of 50 Å^2^ in size) that were potentially involved in the interactions with the hydrophobic moieties of molecules. 

### 2.12. Docking Simulation

Physicochemically and thermodynamically possible binding modes of various small compounds to the Ebola virus glycoprotein model were assessed using the Dock application of MOE as previously described [61]. The heparin, dextran sulfate, and chondroitin sulfate structures were obtained from PubChem [62] (CID numbers of heparin, dextran sulfate, and chondroitin sulfate are 96024437, 7849109, and 47206034, respectively). The structure of lamellarin α 20-sulfate was reported in the previous study [23]. The structures of the entire glycoprotein regions and the small compounds were defined as a receptor and ligand, respectively. Subsequently, possible docking poses between the receptor and ligand were comprehensively searched under conditions that yielded the top 100 docking poses. A timer model of the Ebola virus glycoprotein after 100 ns of MD simulation was used for each docking simulation.

### 2.13. Estimation of Binding Free Energy

The estimation of the binding free energy was carried out using MOE as previously described [61]. The 100 poses that were obtained by the docking simulations were ranked based on the docking scores using the Dock tool in MOE [63], which assesses the steric and thermodynamical validity of the binding poses. The docking poses that were ranked in the top 100 were used to calculate the binding free energies of sulfated compounds to the Ebola virus glycoprotein at pH 7.0 and 310 K using the Potential Energy tool in MOE. In some cases, the binding free energies were calculated after modeling the Ebola virus glycoprotein trimer bound to the sulfated compounds at pH 5.0 and 310 K using the Protonate 3D tool [64] of MOE. The top 100 docking poses were also used to identify the sites of non-covalent interactions between atoms, such as hydrogen bonds, Van der Waals interactions, π-interactions, and ionic interactions.

### 2.14. Estimation of Interaction Sites

The interaction sites between the Ebola virus glycoprotein trimer and sulfated compounds were identified using the Contact Analysis tool in MOE, as previously described [65,66]. This tool allows of the non-covalent interactions that take place between atoms, such as hydrogen bonds, Van der Waals interactions, π-interactions, and ionic interactions to be identified. The frequency of the non-covalent interactions was schematically expressed as barcodes using protein ligand interaction fingerprints (PLIF) analysis. The second-dimension imaging of each binding mode was drawn using the ligand interaction tool.

### 2.15. Shannon Entropy Analysis

Amino acid variation at individual positions of the Ebola virus glycoprotein was quantitated with Shannon entropy as previously described [56,67]. Entropy scores were calculated using a total of 2325 Ebola virus glycoprotein sequences from the Virus Pathogen Resource (ViPR) [68] using Shannon’s equation [69]:$$H\left(i\right)=-\underset{x_{i}}{\sum }p\left(x_{i}\right)\log_{2}p\left(x_{i}\right)\;\left(x_{i}=G,A,I,V,\ldots \right)$$
where *H(i)*, *p(x_i_)*, and *i* indicate the amino acid entropy score of the individual position, the probability of occurrence of a given amino acid at that position, and the number of the position, respectively. An *H(i)* score of zero indicates absolute conservation, whereas a score of 4.4 bits indicates complete randomness.

## 3. Results

### 3.1. Human 293T and TE671 Cells Are Susceptible to Ebola Virus- or SARS-CoV-2-Pseudotyped HIV-1 Vector Infection

African green monkey Vero cells are typically used as target cells for SARS-CoV-2 [3,70,71,72], but they are resistant to HIV-1 vector infection due to the TRIM5α host restriction factor [73,74,75]. Therefore, the susceptibilities of human 293T, HeLa, and TE671 cells to the pseudotyped HIV-1 vector were analyzed. These cell lines are frequently used as target cells for pseudotyped vectors [8,41,76,77].

To obtain the Ebola virus- or SARS-CoV-2-pseudotyped HIV-1 vectors, 293T cells were transfected with HIV-1 Gag-Pol-Tat-Rev and luciferase-encoding HIV-1 vector genome expression plasmids as well as with the Ebola virus or SARS-CoV-2 envelope glycoprotein expression plasmid. Culture supernatants from the transfected cells were used to inoculate 293T, HeLa, and TE671 cells 2 days after the transfection. The luciferase activities of the inoculated cells were measured 2 days after the inoculation. We previously demonstrated that 293T, HeLa, and TE671 cells were all susceptible to VSV-pseudotyped HIV-1 vector infection [33,78]. The luciferase activities of 293T and TE671 cells inoculated with the pseudotyped HIV-1 vectors were clearly higher than those of uninoculated cells independently of the viral envelope glycoproteins (Figure 1A, left panel). The luciferase activity of HeLa cells inoculated with the SARS-CoV-2-pseudotyped vector was similar to that of uninoculated cells, showing that HeLa cells were resistant to SARS-CoV-2 envelope-mediated infection. Although there are several lines of evidence showing that 293T cells are resistant to SARS-CoV-2 [4,79], it has been reported that 293T cells are susceptible [3,4,10,11,80], which is consistent with our present findings. Human 293T cells are much more susceptible to Ebola virus-pseudotyped HIV-1 vector infection than TE671 cells and are widely used as targets for the HIV-1 vector by many other research groups. Although ACE2-expressing 293T cells are frequently used in the SARS-CoV-2 study, the over-expression of the exogenous ACE2 protein may induce unexpected effects. Normal 293T cells were therefore used as target cells in the following experiments.

The luciferase activity of cells inoculated with the VSV-pseudotyped HIV-1 vector was much higher than that of cells inoculated with the Ebola virus or SARS-CoV-2-pseudotyped HIV-1 vector (Figure 1A, right panel). Therefore, culture supernatant containing VSV-pseudotyped HIV-1 vector was diluted 100 times with culture media when it was used as a control.

Human 293T cells have been reported to be less susceptible to infection with SARS-CoV-2 [3,4,10,11,80], which is consistent with our present results in Figure 1A. To confirm whether the luciferase activity in our infectivity assay was generated by viral envelope-mediated infection, we conducted neutralization experiments using antibodies against the envelope glycoproteins of SARS-CoV-2 and Ebola virus. The antibodies effectively blocked the 293T cells from being infected with SARS-CoV-2 and Ebola virus-pseudotyped HIV-1 vectors in a dose–response manner (Figure 1B). The results indicate that the luciferase activity in our infectivity assay indeed reflected viral envelope-mediated infection.

### 3.2. Lamellarin α 20-Sulfate Inhibits Ebola Virus- or SARS-CoV-2-Pseudotyped HIV-1 Vector Infection

Lamellarin α 20-sulfate (Figure 2A) was synthesized as previously described [23]. To examine whether lamellarin α 20-sulfate inhibited Ebola virus-pseudotyped HIV-1 vector infection, target 293T cells were pretreated with lamellarin α 20-sulfate for 5 h and were then inoculated with the pseudotyped HIV-1 vector in the presence of lamellarin α 20-sulfate. Luciferase activity decreased with 150 µM of lamellarin α 20-sulfate (Figure 2B). 

To assess the possibility that the HIV-1 Gag-Pol protein is the target of lamellarin α 20-sulfate, the Ebola virus-pseudotyped murine leukemia virus (MLV) vector was constructed [41]. Human 293T cells were transfected with the MLV Gag-Pol, luciferase-encoding MLV vector genome, and Ebola virus glycoprotein expression plasmids. The culture supernatants of the transfected cells were inoculated to 293T cells in the presence of lamellarin α 20-sulfate (150 µM). The compound decreased luminescence, indicating that lamellarin α 20-sulfate inhibits Ebola virus-pseudotyped MLV vector infection (Appendix A). These results suggest that the HIV-1 Gag and Pol proteins in the HIV-1 vector is not the primary target of lamellarin α 20-sulfate.

To determine whether lamellarin α 20-sulfate also affects cell viability, 293T cells were cultured for 2 days in the presence of lamellarin α 20-sulfate, and the number of live cells was counted. Treatment with lamellarin α 20-sulfate only slightly reduced the number of live cells, and the difference was not statistically significant (Figure 2C), suggesting that the effects of lamellarin α 20-sulfate on cell viability are minimal if there are any at all.

Lamellarin α 20-sulfate reduced the luminescence of SARS-CoV-2-pseudotyped HIV-1 vector-inoculated cells in a dose-dependent manner but did not reduce the number of VSV-pseudotyped HIV-1 vector-inoculated cells (Figure 2D). These results indicate that the inhibitory effects of lamellarin α 20-sulfate is specific to the envelope glycoproteins of Ebola virus and SARS-CoV-2.

### 3.3. Lamellarin α 20-Sulfate Inhibits Ebola Virus Infection by Its Interaction with Viral Glycoprotein

To gain insights into the mechanism(s) by which lamellarin α 20-sulfate inhibits Ebola virus-pseudotyped HIV-1 vector infection, we conducted a time-of-addition experiment using a method reported in a published paper [42]. Target 293T cells were treated with the compound either before or after the inoculation with the vector, as indicated in Figure 3 and in the Materials and Methods section. In the case of pretreated cells, the luminescence was not changed. However, when the 293T cells were cotreated, the luminescence was significantly decreased. When the target cells were treated 5 h after vector inoculation, the level of reduction in the luciferase activity was moderate. If a cellular factor was the target of lamellarin α 20-sulfate, then luminescence should have decreased in the pretreated cells. These results suggest that vector particles are the primary target of lamellarin α 20-sulfate and that the compound inhibits the early stage(s) of Ebola virus infection: vector binding and entry.

### 3.4. Sulfated Polysaccharides Inhibit Ebola Virus- or SARS-CoV-2-Pseudotyped HIV-1 Vector Infection

We have reported that the sulfate residue of lamellarin α 20-sulfate is required for antiviral activity against HIV-1 envelope-mediated infection [23]. Sulfated polysaccharides also inhibit infections caused by HIV-1 [81], herpes simplex virus [29], dengue virus [25], and human papilloma virus [27,82], and their sulfated residues are required for antiviral activities. These facts suggest that lamellarin α 20-sulfate inhibits viral infection via the same mechanism as the one employed by sulfated polysaccharides.

To analyze the impact of sulfated polysaccharides on Ebola virus-, SARS-CoV-2, or VSV-pseudotyped HIV-1 vector infection, 293T cells were pretreated with heparin, dextran sulfate, fucoidan, or chondroitin sulfate and were then inoculated with the pseudotyped HIV-1 vector in the presence of sulfated polysaccharides at the same concentration. The luciferase activities of the SARS-CoV-2-pseudotyped vector-inoculated cells were significantly decreased by all of the sulfated polysaccharides examined in a dose-dependent manner (Figure 4A). Similarly, the luciferase activities of the Ebola virus-pseudotyped vector-inoculated cells were reduced by all of the sulfated polysaccharides that were examined (Figure 4B). 

However, heparin, dextran sulfate, and fucoidan reduced the luciferase activities of the VSV-pseudotyped vector-inoculated cells, but chondroitin sulfate did not (Figure 4C). All of the sulfated polysaccharides did not alter the numbers of live cells (Figure 4D). These results indicate that heparin, dextran sulfate, fucoidan, and chondroitin sulfate inhibit infections that are mediated by the envelope glycoproteins of Ebola virus and SARS-CoV-2 and that the inhibitory effect of chondroitin sulfate is dependent on the viral envelope glycoprotein.

### 3.5. Heparin, Dextran Sulfate, and Fucoidan Inhibit the Interaction of Vector Particles with Target Cells

To examine whether lamellarin α 20-sulfate and sulfated polysaccharides inhibit vector infection by suppressing the interaction of vector particles with target cells, 293T cells were incubated with the pseudotyped HIV-1 vector at 4 °C for 2 h in the presence or absence of lamellarin α 20-sulfate, heparin, dextran sulfate, fucoidan, or chondroitin sulfate and were washed with PBS. The HIV-1 p24 protein levels of the cell lysates prepared from the incubated cells were measured. Because the transduction titer of the VSV-pseudotyped HIV-1 vector was much higher than the other pseudotyped vectors, the VSV-pseudotyped vector was diluted with culture media to normalize the transduction titer. However, the p24 levels of the cell lysate prepared from cells incubated with the VSV-pseudotyped vector was similar to those of the other vectors. Thus, in this experiment, the VSV-pseudotyped vector was not diluted.

Heparin, dextran sulfate, and fucoidan reduced the levels of the p24 protein in the cells incubated with each pseudotyped HIV-1 vector, showing that these compounds inhibit the interaction of the pseudotyped vectors with 293T cells (Figure 5A–C). However, the p24 levels were not changed by lamellarin α 20-sulfate and chondroitin sulfate (Figure 5D,E), which did not inhibit VSV-pseudotyped HIV-1 vector infection. These results indicate that heparin, dextran sulfate, and fucoidan inhibit Ebola virus- or SARS-CoV-2-pseudotyped vector infections by suppressing the interaction of these vectors with target cells. Lamellarin α 20-sulfate and chondroitin sulfate also inhibited infections caused by the envelope glycoproteins of Ebola virus and SARS-CoV-2, but the mechanism underlying these inhibitions was unknown.

### 3.6. Molecular Modeling of the Glycosylated Glycoprotein Trimer of the Reston Ebola Virus Strain

The above results disclose unique characteristics of lamellarin α 20-sulfate and chondroitin sulfate—namely, they display broad-spectrum antiviral activity against various emerging viruses without detectable inhibitory effects on the cell-attachment step. To gain molecular insights into the mechanisms underlying these puzzling phenomena, we conducted an in silico structural study. In this study, we focused on the inhibition mechanisms of Ebola virus because the Ebola virus glycoprotein is relatively rich in information on the structural basis for establishing infection [47,48,83,84,85,86,87,88,89,90,91,92].

To create a basis for the in silico analysis in the present experiments, we first constructed a structural model of the envelope glycoprotein trimer of the Reston Ebola virus strain [43] used in the experiments by means of homology modeling and MD simulation, similar to the preparation for the structure–function analyses of the trimeric envelope proteins of HIV-1 [44,46] and influenza virus [45]. Previously reported information from a high-resolution crystal structure of an unliganded glycoprotein trimer of a Zaire Ebola virus strain, termed GPΔ [47], and on the *N*-glycosylation sites of the Zaire Ebola virus glycoprotein [48] was used to construct the model of the glycoprotein trimer of the Reston Ebola virus strain, which we named GPΔ-Res (Figure 6A,B). The GPΔ and therefore GPΔ-Res lack the mucin-like domain that is nonessential for the viral infection of cells [85,86] in the signal peptide for glycoprotein transport at the late stages of viral replication in the infected cells and in the C-terminal transmembrane region for anchoring in the lipid bilayer of virion particles (Figure 6A). Meanwhile, GPΔ and GPΔ-Res contain an ectodomain that is essential for cell infection [85,86]. Thus, the GPΔ-Res has all of the structural entities that are necessary for the cell attachment and membrane fusion of Ebola virus in the target cells.

The GPΔ-Res was subjected to MD simulations to characterize the structural dynamics of the glycoprotein ectodomain at 1 atm, 310 K, and in 150 mM NaCl (Figure 6B). The dynamics were monitored by the RMSDs between the initial model structure and the structures at given time points in the MD simulation (Figure 6C). The RMSDs sharply increased at the beginning of the MD simulations within a few nanoseconds and then reached a near plateau with continual fluctuations (Figure 6C). The results suggest that structural distortions of the initial model based on the crystal structure were relieved shortly after the start of MD simulation, reaching a state of thermodynamic equilibrium in solution. Consistent with the reported Ebola virus glycoprotein trimer structure [47], the glycan cap of the GP1 subunit had kept in pace to overlay the fusion loop and binding surface of Neimann–Pick C1 (NPC1), the regions critical for the proposed conformational changes of the glycoprotein for membrane fusion in endosomes [87,88,89,90,91,92] (Figure 6D). These results suggest that the ectodomain of the unliganded Ebola virus glycoprotein maintains a fusion-incompetent structure under neutral conditions. 

### 3.7. Characterization of Chemical Features of the Ebola Virus Glycoprotein Surface for the Molecular Interactions

To gain insights into the chemical features of the Ebola virus glycoprotein for molecular interactions, we conducted molecular patch analyses [56] using the GPΔ-Res model. The antiviral activity of sulfated compounds requires a negatively charged sulfate [23,29,30,31,32]. This suggests that the binding of sulfated compounds to the Ebola virus glycoprotein involves the participation of electrostatic interactions. Therefore, we first searched for the positively charged areas of GPΔ-Res that could allow electrostatic interactions with the sulfated compounds using the Protein Patch Analyzer tool in MOE [56]. The positively charged patches with a minimum area of 40 Å^2^ for the electrostatic interactions were distributed throughout the GPΔ-Res trimer structure obtained at 100 ns during the MD simulation (Figure 7A). Notably, some patches were located near the reported functional sites, such as the NPC1-binding surface and fusion loop, in the head region of GPΔ-Res (Figure 6D). 

Lamellarin α 20-sulfate is a polyaromatic alkaloid that is rich in aromatic rings for hydrophobic and π-stacking interactions [23] (Figure 2A). Hydrophobic interactions play important roles in generating the binding specificity of the sulfated polysaccharides to the SARS-CoV-2 spike protein [32] and herpes simplex virus type 1 glycoprotein C [94]. Therefore, we next searched for the hydrophobic patches on the glycoprotein that could support lamellarin α 20-sulfate binding using the MD-based GPΔ-Res model. Hydrophobic patches with a minimum area of 50 Å^2^, which are presumably important during hydrophobic interactions [95,96], were searched for using the Protein Patch Analyzer tool in MOE [56]. Again, the hydrophobic patches were detected near the functional sites of the glycoprotein head region (Figure 6D and Figure 7B) [83,84]. Together, these results suggest that the Ebola virus glycoprotein is rich in sites for electrostatic and hydrophobic interactions with negatively charged molecules with hydrophobic moieties. 

### 3.8. Characterization of Binding Modes of Sulfated Compounds on the Reston Ebola Virus Glycoprotein

The chemical features of the Ebola virus glycoprotein surface alone are insufficient for the binding of sulfated compounds. To clarify the chemically, structurally, and thermodynamically appropriate binding sites of each sulfated compound, we conducted in silico docking simulations with the glycosylated GPΔ-Res trimer structure at 100 ns of MD simulation (Figure 8A). Both commercially available and academic docking programs generally provided good predictions for a set of appropriate ligand binding poses, whereas the ranks of the docking poses based on the binding affinity could not always be predicted well [97]. Therefore, we used the Dock tool in MOE to characterize the top 100 docking poses ranked based on the degrees of binding affinity and steric hindrance. Lamellarin α 20-sulfate and chondroitin sulfate exhibited larger binding free energy values compared to heparin and dextran sulfate in the top 100 poses (Figure 8B), suggesting that the former compounds have smaller binding affinities to the GPΔ-Res. However, the differences in the mean values were not statistically significant. Thus, the binding affinities were basically indistinguishable among the sulfated compounds and unlikely to explain the unique antiviral action of lamellarin α 20-sulfate and chondroitin sulfate against Ebola virus.

To clarify the binding sites for the sulfated compounds, the glycoprotein residues that had been involved in the noncovalent interactions with the compounds more than 10 times in the 100 docking poses were identified using the Contact Analysis tool in MOE (Figure 8C). Interestingly, a marked difference was detected in the distribution of the glycoprotein residues; the residues in the attachment subunit GP1 [47,98] often functioned to support the binding of heparin and dextran sulfate, whereas the GP1 residues rarely functioned to support the binding of chondroitin sulfate and lamellarin α 20-sulfate (Figure 8C). 

These residues constituted the principal binding regions that differentiated the binding mode of each compound (Figure 8D). First, the Ebola virus glycoprotein had a cavity between the G1 and G2 subunits that was specific to heparin, dextran sulfate, and lamellarin α 20-sulfate binding (Figure 8D, side view, red arrows). Second, the glycoprotein had another cavity between the G1 and G2 subunits that was specific to dextran sulfate and chondroitin sulfate binding (Figure 8D, side view, green arrows). Interestingly, this binding site for dextran sulfate and chondroitin sulfate corresponded to one of the three reported binding sites for the already approved, nonsulfated anti-Ebola virus drugs [47,98]. Finally, the large cavity formed by three GP1 subunit monomers was found to possess structural features that are suitable for binding to heparin and dextran sulfate but not to chondroitin sulfate or lamellarin α 20-sulfate (Figure 8D, top view). Thus, the binding sites for chondroitin sulfate and lamellarin α 20-sulfate are basically confined to cavities between the GP1 and GP2 subunits, whereas those for heparin and dextran sulfate are dispersed across the glycoprotein head region and primarily located on the frontal attachment regions of the GP1 subunit. 

### 3.9. Characterization of Molecular Interactions between Sulfated Compounds and Ebola Virus Glycoprotein

To gain insight into the molecular mechanisms underlying antiviral actions, we further characterized the sites and chemical features of the molecular interactions between lamellarin α 20-sulfate and Ebola virus GPΔ-Res. The principal binding site predicted for lamellarin α 20-sulfate (Figure 8D, lamellarin α 20-sulfate) accommodated 32 docking poses that exhibited the best and majority of the higher-level binding affinities among the 100 poses (Figure 9A). This binding site was located near the glycan at N564 in the Reston Ebola virus glycoprotein (N563 for the Zaire Ebola virus glycoprotein [47]) and constituted a hydrophobic cavity with patches of hydrophilic residues, such as S82, R86, and K511 (Figure 9B). Thus, both electrostatic and hydrophobic interactions could play key roles in the binding that takes place at this site. Although the principal binding sites of chondroitin sulfate were different from those for lamellarin α 20-sulfate, the types of interactions were identical (Appendix A).

Notably, the lamellarin binding site was situated near the critical structural units for the conformational glycoprotein changes, i.e., the NPC1 binding surface and fusion loop [87,88,89,90,91,92] (Figure 9C). In the number 1 pose with the highest docking score, lamellarin α 20-sulfate was placed near the α1 helix and α1′ helix, which undergo conformational changes upon receptor binding to confer the glycoprotein with fusion competency [47]. In addition, lamellarin α 20-sulfate formed ionic and hydrogen bonds with K511 at the base of the fusion loop, generating a new noncovalent interaction network between lamellarin, K511, and the α1′ helix (Figure 9C). Importantly, when the binding affinities of individual sulfated compounds were calculated in conditions with a pH level of 5.0, the affinities were all augmented under the acidic conditions (Figure 9D). Finally, two key basic residues for lamellarin binding, R86 and K511, were almost perfectly conserved among the amino acid sequences of the Ebola virus glycoprotein from the public database ViPR [68] (Appendix A, *n* = 2325), suggesting strong structural and/or functional constraints on these residues. 

## 4. Discussion

This study showed that a marine alkaloid, lamellarin α 20-sulfate, and various sulfated polysaccharides, such as heparin, dextran sulfate, fucoidan, and chondroitin sulfate, have a capacity to inhibit the infection of Ebola virus- or SARS-CoV-2-pseudotyped HIV-1 vector. During the course of this study, other groups reported that heparin, fucoidan, and chondroitin sulfate could inhibit the infection of SARS-CoV-2 [32,99,100]. Our results are consistent with these reports and further provide the first evidence that lamellarin α 20-sulfate and dextran sulfate also possess antiviral activity against various emerging RNA viruses that require endocytosis to establish infections. Furthermore, our study provides novel information on the unique biological and physicochemical features of lamellarin α 20-sulfate. To our knowledge, this is the first report to address the possibility of an unreported point of action of the marine alkaloid against enveloped RNA viruses.

Using pseudotyped viruses, we have demonstrated that lamellarin α 20-sulfate could inhibit viral infections that are mediated by the glycoproteins of Ebola virus and SARS-CoV-2 without suppressing virion attachment to susceptible cells (Figure 2 and Figure 5). The sulfated polysaccharides and marine alkaloids that were tested inhibited the infection of both MLV and HIV vectors with the Ebola virus envelope protein (Figure 4B and Appendix A). In addition, the pretreatment of lamellarin α 20-sulfate did not inhibit HIV–vector infection (Figure 3). These results strongly suggest that the primary targets of these inhibitors are the Ebola virus envelope protein rather than other viral and cellular proteins. The results also indicate that the antiviral activities of lamellarin α 20-sulfate against these RNA viruses take place after the attachment of the virus to the target cells. Since VSV-G-mediated infection requires endocytosis and endosome acidification [33,41], and since lamellarin α 20-sulfate did not inhibit VSV infection (Figure 2B), the antiviral action is unlikely to include the dysfunction of endocytosis and endosome acidification of the cells. Likewise, lamellarin α 20-sulfate exhibited a dose-dependent inhibition against SARS-CoV-2 under the minimum cell toxicity (Figure 2C), suggesting that the main binding target of this compound is unlikely to be the cellular proteins involved in the maintenance of cellular metabolism. In this regard, it should be noted that lamellarin α 20-sulfate is a cell-membrane-impermeable compound [23] and should rarely, if ever, infiltrate into the interiors of cells and virions. Together, the present and previous results consistently indicate that the primary targets of lamellarin α 20-sulfate against the viruses tested are the small-compound accessible regions of the viral glycoproteins on the virions, whereas the antiviral actions somehow take place after the cell-attachment and endocytosis of the virions.

Our in silico structural analyses strongly support the above possibility derived from the conducted experiments. A physicochemically and thermodynamically favorable binding site existed in the Ebola virus glycoprotein and was confined to a region that was situated on the side of the glycoprotein rather than the frontal regions for viral attachment (Figure 8D). The principal binding site of lamellarin α 20-sulfate was found to constitute a previously unappreciated hydrophobic cavity near the NPC1-binding site and the base of the fusion loop on the trimeric glycoprotein of the Ebola virus (Figure 9). Therefore, it is possible that the lamellarin binding to this site caused an alteration in the fluctuation of the NPC1-binding surface and thereby changed the binding affinity to NPC1. Furthermore, the binding of lamellarin α 20-sulfate to the base of the fusion loop would interfere with the functional conformational changes of the loop and/or GP2 subunit to create a fusion-competent structure. Notably, lamellarin binding to this cavity could be augmented at pH 5.0 (Figure 9D), suggesting that the compound could continue to bind to the glycoprotein under acidic conditions. These in silico and experimental results are consistent with the assumption that lamellarin α 20-sulfate binds to the glycoprotein on the virion outside the cells and then functions after virion internalization as an inhibitor against conformational changes in the glycoprotein for genome uncoating in endosomes.

Our in silico and experimental results indicate that chondroitin sulfate exhibits antiviral activity at stages similar to those of lamellarin α 20-sulfate. This compound could inhibit the virus infections of various viruses without suppressing virion attachment to the cells (Figure 2 and Figure 5). Consistent with this idea, we identified no preferential binding sites on the frontal regions of the glycoprotein (Figure 8D). On the other hand, we identified a preferential binding site in a hydrophobic cavity near the fusion loop (Appendix A), and binding to this cavity could be augmented under the pH 5.0 condition (Figure 9D). Therefore, although the principal binding site probably differs between lamellarin α 20-sulfate and chondroitin sulfate, both compounds seem to work in endosomes.

It is curious that chondroitin sulfate can inhibit the binding of the envelope glycoprotein of HIV-1 and dengue virus to their receptors on the cell surface [101,102,103]. Likewise, we found that lamellarin α 20-sulfate could inhibit cell–cell fusion [41], which is mediated by the HIV-1 envelope glycoprotein and cell-surface CD4 [104]. These observations suggest that the points of antiviral actions by these compounds could be the cell-attachment step. Thus, the inhibition steps of these compounds differ from virus to virus, perhaps according to the individual strategies of infection and the structures of the viral envelope glycoproteins. The present study suggests that the primary points of the antiviral actions against viruses that use endocytosis are the events occurring after the entry of these compounds and possibly the events in endosomes.

In contrast to these two compounds, sulfated polysaccharides, such as heparin and dextran sulfate, are likely to have physicochemical features that allow them to work at both the cell-attachment and conformational change stages in endosomes. We found that the sulfated polysaccharides could inhibit viral attachment on the target cells (Figure 5), that they had preferential binding sites both on the frontal and side regions of the glycoprotein (Figure 8D), and that binding to these cavities could be augmented under the pH 5.0 condition (Figure 9D). In this regard, it should be noted that SARS-CoV-2 infection depends on the cell-surface heparan sulfate and angiotensin-converting enzyme 2 (ACE2) [32] and that many viral infections are inhibited by soluble sulfated polysaccharides [31,100]. These observations may imply that the viruses inhibited by these sulfated polysaccharides utilize cellular heparan sulfate as an “attachment receptor”. Candidate sites of virus binding of such an attachment receptor may include the regions localized on the frontal region of the G1 subunit that were specific to the binding of heparin and dextran sulfate (Figure 8D top view).

The sulfated polysaccharides, heparin, dextran sulfate, and fucoidan, but not chondroitin sulfate and lamellarin α 20-sulfate, inhibited VSV-pseudotyped vector infection (Figure 2D and Figure 4C). The results may reflect the higher binding specificity of the polysaccharides compared to the marine alkaloids. This could be attained by the physicochemical differences in the inhibitors; the polysaccharides are more heavily charged and larger compared to the marine alkaloids. Consistently, the in silico docking study showed that the heparin and dextran sulfate could bind to the multiple sites dispersed across the head region of the Ebola virus glycoprotein (Figure 8D, heparin and dextran sulfate). In marked contrast, potential binding sites were limited in the particular cavities between the GP1 and GP2 subunits for the marine alkaloids (Figure 8D, chondroitin sulfate and lamellarin α 20-sulfate). Thus, the marine alkaloids are likely to require cavities with stricter shapes and with particular chemical properties on the protein, as exampled in the molecular interactions between lamellarin α 20-sulfate and the Ebola virus glycoprotein (Figure 9). Further study is necessary to address each of these issues.

Heparin, dextran sulfate, and chondroitin sulfate are clinically used to treat blood clots, hypertriglyceridemia, and joint pain, respectively. Dextran sulfate is orally administered and inhibits the absorption of glyceride in digestive tissues. In contrast, heparin and chondroitin sulfate can be intravenously administered. Thus, these drugs can be available for COVID-19 immediately. Patients infected with SARS-CoV-2 frequently develop thrombosis, and heparin has been administered to suppress thrombosis in such cases. Heparin treatment was shown to have significantly improved the status of COVID-19 patients [105]. Our study raises a possibility that this improvement was partly attained by inhibiting SARS-CoV-2 replication in those patients.

Lamellarin α 20-sulfate has been shown to be a potent host defense factor in some invertebrates and has been isolated from ascidians (tunicates) [19,22]. These animals do not have adaptive immunity and thus are only able to protect themselves from viruses via innate immunity. It is thought that ascidians utilize lamellarin α 20-sulfate to restrict various viral infections. Indeed, we have shown that this alkaloid has antiviral activities against a broad range of enveloped viruses [23] (Figure 2 and Figure 4). Residues of the Ebola virus glycoprotein key for the binding of lamellarin α 20-sulfate were shown to be highly conserved among various Ebola viruses (Appendix A). Lamellarin α 20-sulfate is present in edible tunicates such as sea squirts, and thus, the compound should have no toxicity for humans. Collectively, the available information suggests that lamellarin α 20-sulfate could be a leading compound for the development of new substances possessing pan-enveloped-virus antiviral activity with minimum side effects. The findings of this study reveal key features of marine alkaloids that could be applied for the further exploration of broadly reactive antivirals on the basis of the structural mechanisms of virus replication.

## Figures and Tables

**Figure 1 viruses-14-00816-f001:**
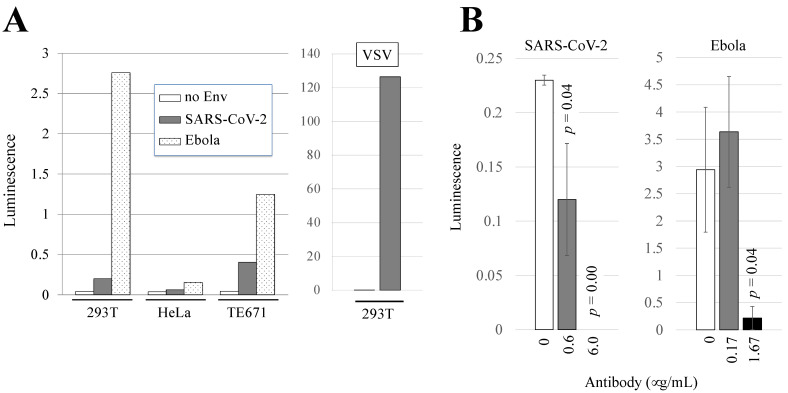
Susceptibility of 293T, HeLa, and TE671 cells to Ebola virus- or SARS-CoV-2-pseudotyped HIV-1 vector infection. (**A**) Human 293T, HeLa, and TE671 cells were inoculated with HIV-1 vector containing Ebola virus or SARS-CoV-2 envelope glycoprotein. The vector encodes the luciferase gene, and luciferase activities of the inoculated cells were measured. This experiment was repeated three times. Representative luminescence values are indicated. (**B**) Target 293T cells were inoculated with indicated pseudotyped HIV-1 vector in the presence of an antibody. Luminescences in inoculated cells were adjusted by the subtraction of those with HIV-1 vector without any viral envelope proteins. The subtracted values are indicated. This experiment was performed in triplicate. Error bars indicate SDs. When the differences compared to control cells are significant, the *p* values of Student’s *t*-test are indicated.

**Figure 2 viruses-14-00816-f002:**
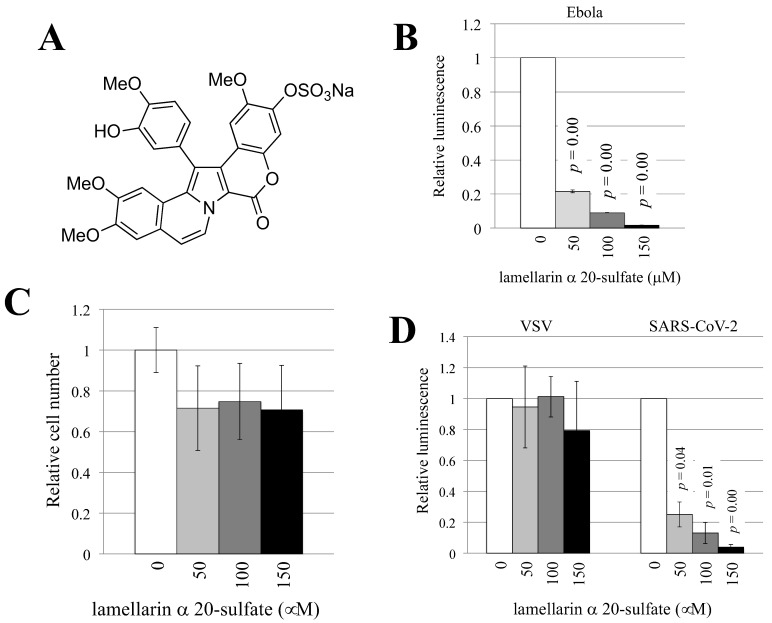
Lamellarin α 20-sulfate inhibits SARS-CoV-2- and Ebola virus-pseudotyped HIV-1 vector infections. (**A**) The chemical structure of lamellarin α 20-sulfate is indicated. (**B**) Human 293T cells were pretreated with DMSO or lamellarin α 20-sulfate at the concentration of 150 µM for 5 h and then inoculated with Ebola virus-pseudotyped HIV-1 vector in the presence of the same concentration of the chemical. The luciferase activities of untreated cells were set to 1, and relative values to the luciferase activities of untreated cells are indicated. This experiment was repeated three times. (**C**) Human 293T cells were cultured for 2 days in the presence of lamellarin α 20-sulfate and stained with trypan blue. The unstained live cells were counted. The average cell number of DMSO-treated cells was set to 1, and the relative values to the number of live cells in DMSO-treated cells are indicated. This experiment was repeated three times. (**D**) Human 293T cells were pretreated with lamellarin α 20-sulfate for 5 h and inoculated with HIV-1 vector containing the indicated envelope glycoprotein in the presence of lamellarin α 20-sulfate. The luciferase activities of the DMSO-treated cells were always set to 1, and the values relative to the luciferase activities of untreated cells are indicated. This experiment was repeated three times. Error bars indicate SDs. When the differences compared to the control cells are significant, the *p* values of Student’s *t*-test are indicated.

**Figure 3 viruses-14-00816-f003:**
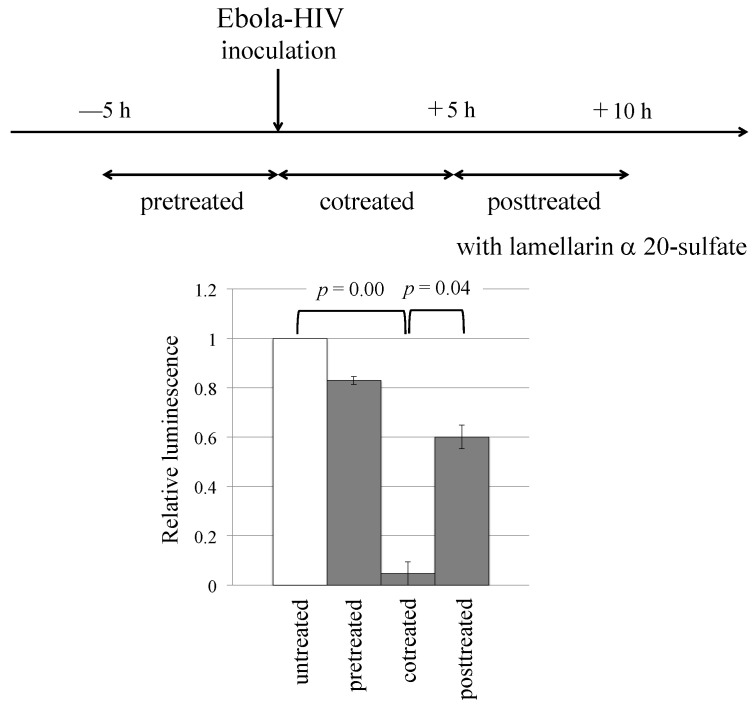
Effects of time at which lamellarin α 20-sulfate was added on inhibitory effects. The 293T cells were treated with lamellarin α 20-sulfate (150 µM) at the time points indicated in the upper panel, as indicated in the Materials and Methods section. The luciferase activities of DMSO-treated cells were always set to 1, and the relative values to the luciferase activities of the DMSO-treated cells are indicated. This experiment was repeated three times. Error bars indicate SD. When the differences between indicated groups are significant, the *p* values of Student’s *t*-test are indicated.

**Figure 4 viruses-14-00816-f004:**
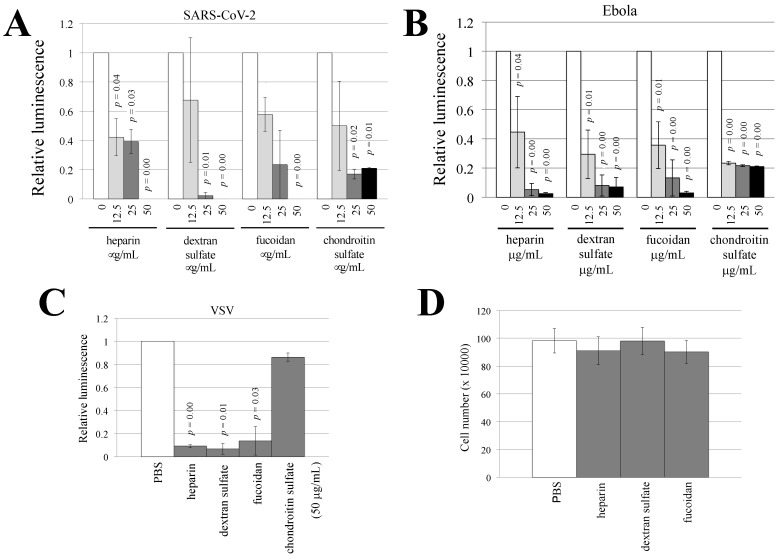
Sulfated polysaccharides inhibit pseudotyped HIV-1 vector infection. (**A**) Human 293T cells were pretreated with the indicated sulfated polysaccharide for 5 h and inoculated with the SARS-CoV-2-pseudotyped HIV-1 vector. The luciferase activities of the PBS-treated cells were always set to 1, and the relative values to the luciferase activities of the PBS-treated cells are indicated. This experiment was repeated three times. (**B**,**C**) Human 293T cells were pretreated with the indicated polysaccharides and inoculated with Ebola virus-(B) or VSV-(C) pseudotyped HIV-1 vector. (**D**) Human 293T cells were treated with the indicated polysaccharides for 2 days and stained with trypan blue. The unstained live cells were counted. This experiment was repeated three times. Error bars indicate SD. When the differences compared to the control cells are significant, the *p* values of Student’s *t*-test are indicated.

**Figure 5 viruses-14-00816-f005:**
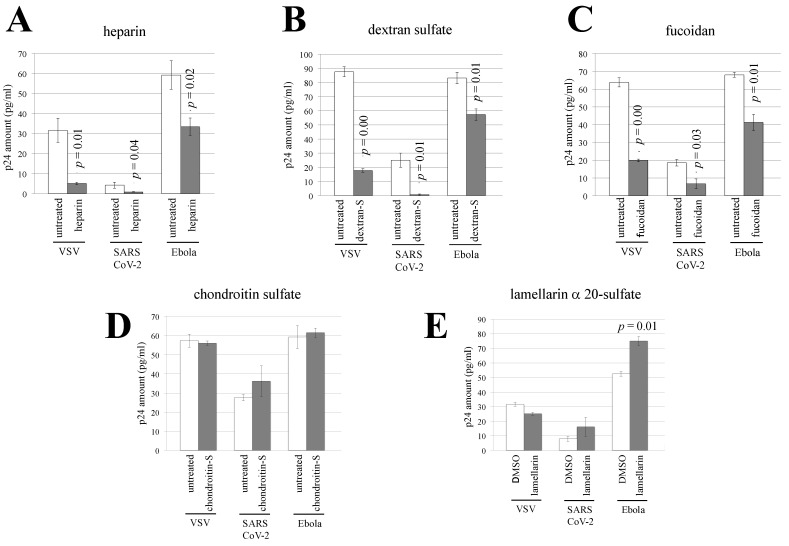
Effects of lamellarin α 20-sulfate and sulfated polysaccharides on the interaction between HIV-1 vector particles and target cells. Human 293T cells were incubated with the indicated pseudotyped vector at 4 °C for 2 h in the presence of heparin (50 µg/mL) (**A**), dextran sulfate (50 µg/mL) (**B**), fucoidan (50 µg/mL) (**C**), chondroitin sulfate (50 µg/mL) (**D**), and lamellarin α 20-sulfate (150 µM) (**E**). The cells were vigorously washed with PBS, and cell lysates were prepared. The amounts of HIV-1 p24 protein in the cell lysates were measured by ELISA. This experiment was performed in triplicate. Error bars indicate SD. When the differences compared to the control cells are significant, the *p* values of Student’s *t*-test are indicated.

**Figure 6 viruses-14-00816-f006:**
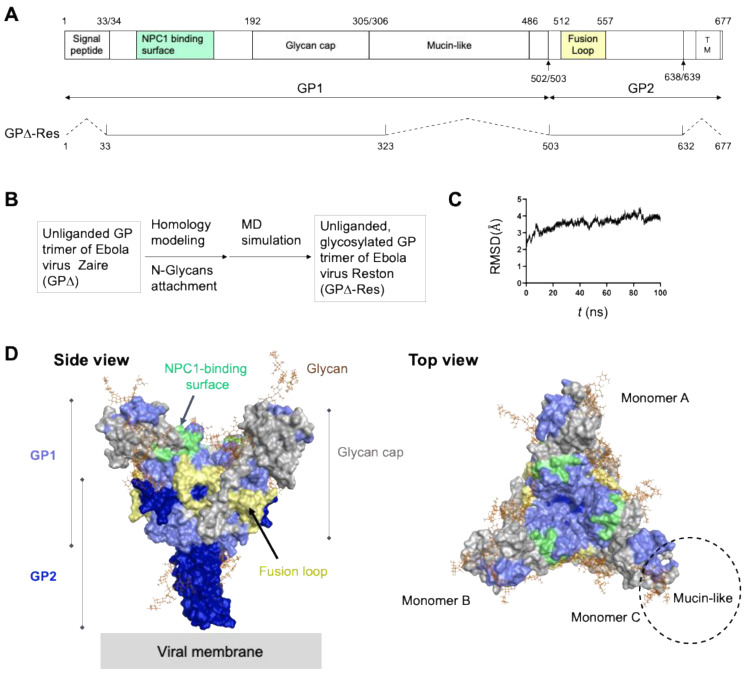
Molecular modeling of the unliganded, pre-fusion state of the glycosylated GPΔ-Res trimer of the Reston Ebola virus strain. The reported amino acid sequence of the spike glycoprotein of the Reston Ebola virus strain (GenBank accession number: AAC54885) [43] and the reported crystal structure of the ectodomain of the unliganded glycoprotein trimer of the Zaire Ebola virus strain at a resolution of 2.23 Å (PDB code: 5JQ3) [47] were used to construct a structural model of the GPΔ-Res trimer of the Reston Ebola virus strain by homology modeling followed by MD simulation, as described for the modeling of the trimeric envelope glycoproteins of other RNA viruses [44,45,46]. Gal_2_Man_3_GlcNAc_4_ was added on the trimeric glycoprotein model before MD simulation based on previously reported information [48] using the Online Glycoprotein Builder web-tool GLYCAM-Web [49]. (**A**) Schematic of the structural organization of the Ebola virus glycoprotein. Dotted lines indicate regions absent in the modeling template GPΔ [47] and the Reston Ebola virus glycoprotein model (GPΔ-Res). Amino acid numbering is based on the Reston glycoprotein. (**B**) Overview of the GPΔ-Res modeling. (**C**) Root mean square deviations (RMSDs) between the initial model structure and the structures at given time points during the MD simulation. (**D**) Molecular surface model of a GPΔ-Res structure at 100 ns of MD simulations. Mucin-like domain is located on the frontal region of glycan cap of each glycoprotein monomer [93]. Light green and yellow portions indicate the binding surface of endosomal protein Neimann–Pick C1 (NPC1) and fusion loop, respectively. These structural units are proposed to be critical for the conformational changes and membrane fusion of the glycoproteins in endosomes [87,88,89,90,91,92].

**Figure 7 viruses-14-00816-f007:**
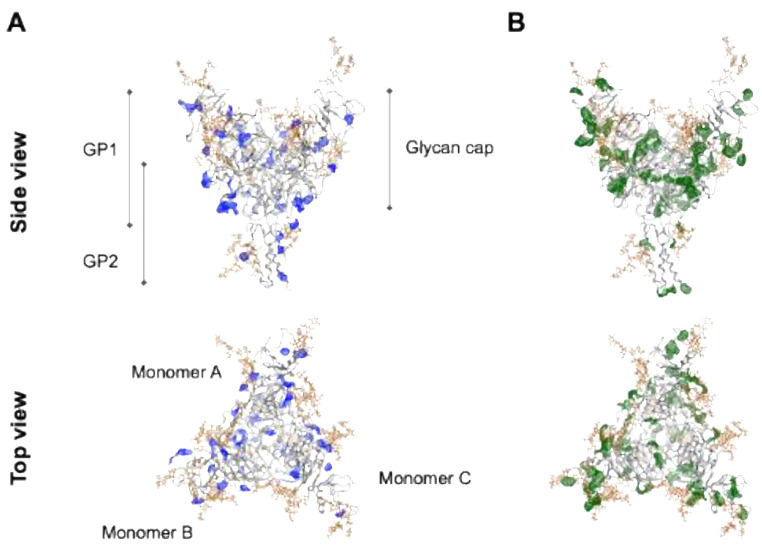
Characterization of chemical features of the Ebola virus glycoprotein surface for the molecular interactions. An Ebola virus GPΔ-Res structure at 100 ns of MD simulations was used to characterize the chemical features of the protein surfaces for molecular interactions using the Protein Patch Analyzer tool in MOE [56,57,58,59,60]. This program calculates electrostatic or hydrophobic patches on the protein to elucidate the 3D distributions of interaction-prone areas. (**A**) Molecular patches relevant to electrostatic interactions. Blue portions in the GPΔ-Res structure indicate the positively charged patches with a minimum area of 40 Å^2^ that potentially interact with negatively charged molecules. (**B**) Molecular patches relevant to hydrophobic interactions. Green portions indicate the hydrophobic patches with a minimum area of 50 Å^2^ that were potentially involved in the interactions with the hydrophobic moieties of molecules.

**Figure 8 viruses-14-00816-f008:**
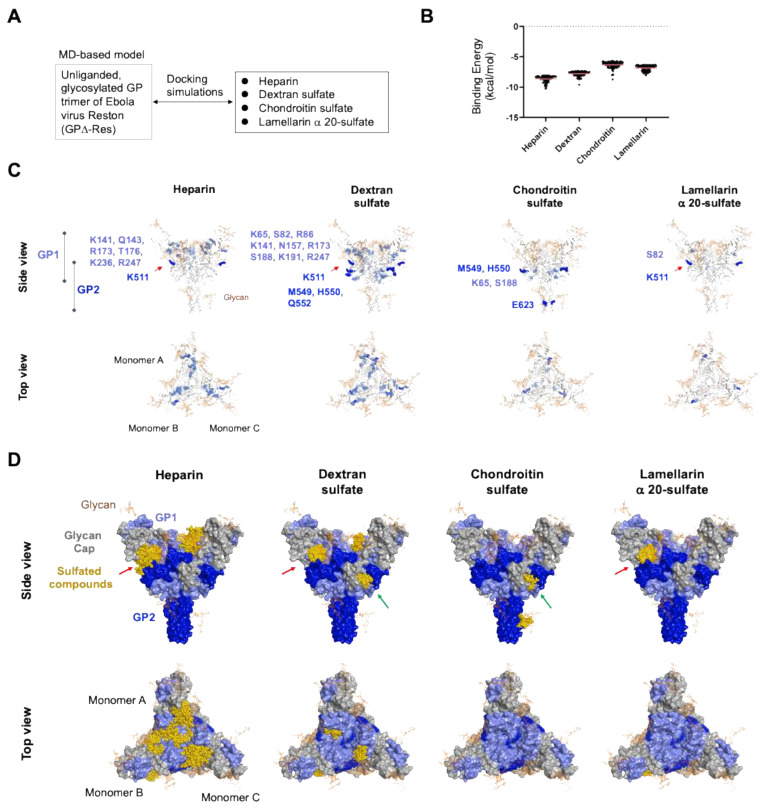
Characterization of possible binding modes of sulfated compounds on the Reston Ebola virus glycoprotein. Docking simulations of sulfated compounds to the Ebola virus GPΔ-Res structure at 100 ns of MD simulation were performed using the Dock application in MOE. (**A**) Overview of docking simulations. (**B**) Distribution of binding energies of the top 100 binding poses. Binding free energies were calculated for each docking pose using the Dock tool in MOE. Red lines indicate mean values. (**C**) Distribution of glycoprotein residues for binding with sulfated compounds. Amino acid residues interacting noncovalently with sulfated compounds were identified with the 100 docking poses using the Contact Analysis tool in MOE. Residues that were involved in the noncovalent interactions more than 10 times in the 100 docking poses are highlighted in different colors. Light blue and blue spheres indicate residues in the GP1 and GP2 subunits, respectively. (**D**) Docking poses of sulfated compounds. Dark yellow spheres indicate atoms of compounds noncovalently bound with the glycoprotein residues in (**C**). Red arrows indicate a binding site common to heparin, dextran sulfate, and lamellarin α 20-sulfate. Green arrows indicate a binding site common to dextran sulfate and chondroitin sulfate, which was found to be identical to a binding site of the approved anti-Ebola virus drugs [47,98].

**Figure 9 viruses-14-00816-f009:**
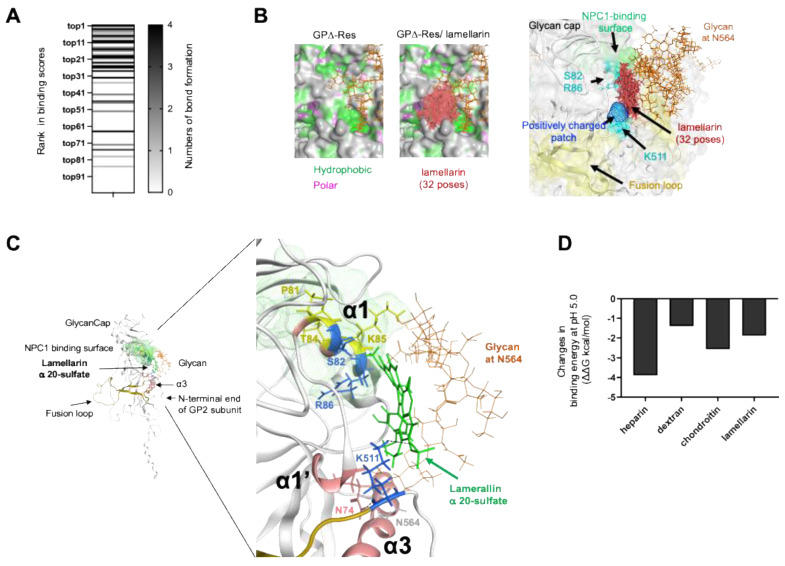
Characterization of molecular interactions between lamellarin α 20-sulfate and the Ebola virus glycoprotein at the major binding sites. (**A**) Ranking of the 32 docking poses in the principal binding site. Individual docking poses of lamellarin α 20-sulfate in the site depicted by the red arrow in Figure 7D are illustrated with horizontal lines along with information about the ranking of the docking scores (letters at left) and the number of noncovalent bonds (dark band at right). (**B**) Superposition of the 32 docking poses of lamellarin α 20-sulfate in the principal binding site. Red rod-like chains indicate lamellarin α 20-sulfate atoms. (**C**) Binding mode with the highest (number 1) docking score. Blue residues in the α1 helix of the GP1 subunit (S82 and R86) and at the base of the fusion loop of the GP2 subunit (K511) indicate those involved in noncovalent interactions with lamellarin α 20-sulfate. Yellow residues in the α1 helix (P81, T84, and K85) are the residues supporting NPC1 binding in the Zaire Ebola virus [91]. The dotted line indicates a hydrogen bond between the main chain of K511 and the side chain of N74. (**D**) Augmentation of the binding affinity of sulfated compounds under acidic conditions. Docking poses with the highest (number 1) scores were used to calculate the binding free energies of sulfated compounds on Ebola virus GPΔ-Res at pH 7.0 and pH 5.0 using the Protonate 3D tool [64] in MOE. Changes in the binding free energies at pH 5.0 are shown.

## Data Availability

Please contact Y.K. and H.S. for the experimental and in silico data, respectively.

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
