# Peer review of "Unique Mode of Antiviral Action of a Marine Alkaloid against Ebola Virus and SARS-CoV-2"

_viruses, 2022, doi:10.3390/v14040816_

Round 1
Reviewer 1 Report
The authors cleary improved the quality of the manuscript, and I recommend it for publication in the current form.
Reviewer 2 Report
The authors have done a very good job in revising the manuscript and incorporating the suggestions from the reviewers. I still have few points which should be considered:
1) Fig 1A TE671 seems better than 293T in supporting SARS-CoV-2-HIV entry and are not bad for Ebola-HIV. However the author chose the 293T. It is not clear why
2)majority of the SARS-CoV-2 paper using SARS-CoV-2HIV use ACE2-transfected 293T or even ACE2 and TMPRSS2 transfected 293T to have a better titer. Why the same has not been done here?
Fig 1B and Fig 4 why for the Ebola-HIV the compounds are used at 1 single dose while for the SARS-CoV-2 there is a nice dose-response curve?
the paragraph lines 389-397 is confusing. The author stated that they do NOT dilute the VSV-HIV stock as done in the infectivity study because VSV-HIV has a similar level of p24. Surely, if in their cell lysate they detect only p24 inside the cells as VSV-HIV is more infectious there will be more p24; indeed towards the end of the paragraph the authors stated that there is more p24 for the Ebola-HIV than SARS-HIV because " 293T cells were more susceptible to Ebola virus-pseudotyped vector infection than SARS-CoV-2-
pseudotyped vector infection"
In their reply to my suggestion to corroborate the pseudotyped data with live neutralisation assay, the authors have stated that the it is difficult to perform this for Ebola as hazard group 4 virus. However, I still think this should be possible for SARS-Cov-2 which is an hazard group 3. Despite not be essential for the acceptance of this manuscript, I still believe it will be a very strong experiment in support of their conclusions.
A couple of points in the Introduction:
line 50. the largest Ebola outbreak recorded was in Western Africa (Guinea/Sierra Leone) not in DRC. The reference is for a clinical trial for the vaccine not for the statement just written.
line 67 p4 to indicate a BSL4 laboratory is an americanism.
Author Response
please see attachment.

This manuscript is a resubmission of an earlier submission. The following is a list of the peer review reports and author responses from that submission.
Round 1
Reviewer 1 Report
the manuscript by Izumida and co-author presents some challenges. The title suggest that the main subject is the role of lamellarin alpha 20-sulphate but there are lots of data mainly confirmatory from the literature on the effect of other molecules (heparin, dextran sulphate, fucoidan and chondroitin) and comparison on the mechanism of action. While the data are interesting, it complicates the message. Are all these molecules required to support the authors' theory? can the important ones being picked and choose to make the storyline more straightforward and the extra data put in the supplementary data?
the other issue is the use of a lentiviral vector for the study. Lamellarin alpha 20-sulphate was first described as an HIV integrase inhibitor. Authors have disagreed with this mechanism of action as they have shown in a previous paper that the molecule does not enter the cells and therefore has to act at the entry level. Without reviewing both sides, the most safe approach would have been to use a different vector; indeed the authors have already developed a MLV-based vector in their previous publication [64]. Also, all the envelopes used pseudotyped quite nicely on a VSV-vector system. I would be more convinced if the same data can be reproduced on a different vector. Finally, SARS-CoV-2 is an hazard group 3 agent; the authors should prove their findings with a real virus experiment, by collaborating with somebody who can performed infectivity assay in BSL3 lab.
The authors used VSV-pseudotyped vector as negative control, however the titer of this PV is not included in the figure 1 and in figure 2 they have not provided the information of how much virus they have input in their experiment. VSV-pseudotyped have a much higher titer than other pseudotyped. In our hands, at least 1000x or more than SARS and Ebola. If the input virus is not normalised, then the results from Figure 2 are misleading because it will required much more drug to reduce the infectivity of the VSV.
Choice of target cell line: 293T- authors claimed that 293T are susceptible to SARS-CoV-2 and they provide 5 references. In reference 1, 2 and 8 there is some low level of infectivity shown, but when the authors have to use 293T for downstream experiments the ACE2 receptor is transfected to allow for an usable titre. The other 2 references cited (9 and 68) exclusively used 293T after overexpression of ACE2.
Also, I would like to know which kit for detection of the luminescence is used. In the M&M is stated Promega, but the reading of luminescence (y axis figure 1) are on a scale with does not fit with other publications using a promega luciferase assay system.
Figure 3c, three of the compounds used inhibit all the pseudotyped vectors including the VSV. How do the author conclude that this is envelope dependent and not vector dependent as everything gets hit?
I have some concern about the references. I haven't checked all of them as there are 91, but some are wrong. I have already mentioned those referring to the 293T permissiveness. Also, line 55, there isn't a reference for Lassa. Line 573-4 "we found that lamellarin alpha 20-sulfate could inhibit cell-cell fusion [64]" there is no mention of lamellarin in that paper .
Reviewer 2 Report
The manuscript focuses on lamellarin α 20-sulfate, a marine alkaloid already reported to be able to suppress infections mediated by the HIV-1 envelope glycoprotein. The authors demonstrated the antiviral ability of lamellarin also against other emerging enveloped RNA viruses that use endocytosis for infection, such as Ebola, Lassa, and SARS-CoV-2 viruses. The article was well organized and the English language is quite good. There are some point I would discuss:
- some references are missing (lines 45-52);
- Suddendly, starting from COVID-19 pandemic, the authors move to Ebola and Lassa infections. Here a link is mandatory;
- Materials and methods section is poor. All materials used codes are missing, for example ATCC code cell lines, lamellarin code...;
- I don't see a significative replication of SARS-CoV-2 in 293T cells. So, all the subsequent results obtained can't be justified;
- To understand better the antiviral potential of lamellarin, other assays shoud be performed, as attachment and entry assays. Please refer to the article doi:10.3390/v13071263.